# Stability of iron-bearing carbonates in the deep Earth's interior

Valerio Cerantola[1,2], Elena Bykova[2,3], Ilya Kupenko[1,†], Marco Merlini[4], Leyla Ismailova[5], Catherine McCammon[2], Maxim Bykov[2,6], Alexandr I. Chumakov[1], Sylvain Petitgirard[2], Innokenty Kantor[1,†], Volodymyr Svitlyk[1], Jeroen Jacobs[1], Michael Hanfland[1], Mohamed Mezouar[1], Clemens Prescher[7], Rudolf Rüffer[1], Vitali B. Prakapenka[8] & Leonid Dubrovinsky[2]

The presence of carbonates in inclusions in diamonds coming from depths exceeding 670 km are obvious evidence that carbonates exist in the Earth's lower mantle. However, their range of stability, crystal structures and the thermodynamic conditions of the decarbonation processes remain poorly constrained. Here we investigate the behaviour of pure iron carbonate at pressures over 100 GPa and temperatures over 2,500 K using single-crystal X-ray diffraction and Mössbauer spectroscopy in laser-heated diamond anvil cells. On heating to temperatures of the Earth's geotherm at pressures to ∼50 GPa $FeCO_3$ partially dissociates to form various iron oxides. At higher pressures $FeCO_3$ forms two new structures— tetrairon(III) orthocarbonate $Fe_4^{3+}C_3O_{12}$, and diiron(II) diiron(III) tetracarbonate $Fe_2^{2+}Fe_2^{3+}C_4O_{13}$, both phases containing $CO_4$ tetrahedra. $Fe_4C_4O_{13}$ is stable at conditions along the entire geotherm to depths of at least 2,500 km, thus demonstrating that self-oxidation-reduction reactions can preserve carbonates in the Earth's lower mantle.

[1] European Synchrotron Radiation Facility, 71, avenue des Martyrs, CS 40220, 38043 Grenoble Cedex 9, France. [2] Bayerisches Geoinstitut, Universität Bayreuth, D-95440 Bayreuth, Germany. [3] P02.2 Extreme Conditions Beamline, Deutsches Elektronen-Synchrotron, Notkestrasse 85, D-22607 Hamburg, Germany. [4] Dipartimento di Scienze della Terra, Università degli Studi di Milano, Via Botticelli 23, I-20133 Milano, Italy. [5] Skolkovo Institute of Science and Technology, Center for Hydrocarbon recovery, Skolkovo Innovation Center, 3, Moscow 143026, Russia. [6] Material Modeling and Development Laboratory, National University of Science and Technology MSIS, Moscow 119049, Russia. [7] Institute of Geology and Mineralogy, Universität zu Köln, Greinstraße 4-6, D-50939 Köln, Germany. [8] GSECARS, Center for Advanced Radiation Sources, University of Chicago, Chicago, Illinois 60437, USA. † Present addresses: Institut für Mineralogie, Universität Münster, Corrensstraße 24, D-48149 Münster, Germany (I.Ku); MAX IV Laboratory, Fotongatan 2, 225 94 Lund, Sweden (I.Ka). Correspondence and requests for materials should be addressed to V.C. (email: valerio.cerantola@gmail.com) or to E.B. (email: elena.bykova@desy.de) or to L.D. (email: Leonid.Dubrovinsky@Uni-Bayreuth.DE).

Plate tectonics drives subduction of carbonate-bearing oceanic plates, that are responsible for recycling carbon from the surface down to the deepest regions of our planet. Indeed, geophysical, geochemical and petrological evidence[1–4] suggest that sufficiently cold and/or fast subducting slabs can penetrate the transition zone and the Earth's lower mantle, possibly even reaching the core–mantle boundary. Subducting plates are the major source of carbon influx inside the Earth[5]. Observation of carbonate inclusions in super-deep diamonds of lower mantle origin is evidence for their existence at depths greater than 700 km (refs 6–8). Untangling the behaviour of carbonates at extreme conditions, that is, determining their stability regions and properties, is a key to understanding the deep carbon cycle.

There are two major mechanisms that could affect carbonate phase stability and carbon oxidation state in the Earth's interior—chemical reaction(s) with surrounding minerals or transformations (including self-oxidation-reduction) of carbonates themselves at specific pressures and temperatures. Previous studies on the Ca, Mg, Fe-bearing carbonates have established that they all undergo several high-pressure high-temperature (HPHT) phase transitions without decomposing in the pressure range up to 140 GPa and restricted temperatures[9–13]. Investigations of the stability of $MgCO_3$ in the transition zone and upper part of the lower mantle as a function of oxygen fugacity demonstrated that carbon is expected to occur as diamond and carbides in the bulk mantle (when homogenously distributed) rather than carbonates[14]. However, in subducting slabs carbonates are expected to be stable due to the more oxidizing conditions compared to the surrounding mantle[15], which may preserve them to the bottom of the lower mantle. The presence of iron is crucial to the fate of high-temperature carbonates[13,16]. Iron can radically change the thermodynamic stability of carbonate phases, thereby preserving them from breaking down. This behaviour may be a direct consequence of pressure-induced spin crossover[17–21], which has been observed to occur at ~43 GPa at room temperature to over 50 GPa at ~1,200 K (ref. 22) for the endmember $FeCO_3$. The presence of Fe-bearing carbonates in the lower mantle is supported by experimental evidence[12,13]. Iron plays a fundamental role in the redox state of the mantle[23] due to its ability to exist in multiple valence states, and its abundance in the mantle is sufficient to govern the redox state of other elements, carbon in particular.

Interest in the high-pressure behaviour of carbonates has been enhanced by recent reports of novel compounds containing tetrahedral $CO_4^{4-}$ groups instead of the triangular planar $CO_3^{2-}$ groups that occur at ambient pressure[9,12,24,25]. Theoretical predictions indicate potential analogues between $CO_4$-bearing carbonates and silicates[25], but so far experimental information about structures of high-pressure carbonates are too limited (and indeed controversial) to speculate about their crystal chemistry.

In this study, we performed an experimental investigation of the high-pressure high-temperature behaviour of synthetic iron carbonate ($FeCO_3$). Experimental conditions of our work cover the entire mantle and reveal two novel compounds containing tetrahedral $CO_4$ groups, as well as the complex role of ferrous and ferric iron in stabilizing carbonates at extreme conditions. Our single-crystal X-ray diffraction data unambiguously establish the existence of at least one carbonate with a unique structural type (not known for silicates or other tetrahedral anion-bearing compounds), and demonstrate that the conditions in the Earth's lower mantle do not lead to full decomposition of Fe-based carbonates due to self-oxidation-reduction reaction(s).

## Results

**Synthesis and structures of $CO_4$-bearing Fe-carbonates.** Synthesis of $FeCO_3$ single crystals and their characterization at ambient conditions was described by Cerantola et al.[20]. HPHT experiments were performed in laser-heated diamond anvil cells (DACs) (see Methods section for details). We employed single-crystal X-ray diffraction as the primary method for sample characterization, and powder X-ray diffraction when analysis of single-crystal data were not possible. We used energy-domain Mössbauer spectroscopy (synchrotron Mössbauer source, SMS; see Methods section for details) as a complementary method of phase analysis and to determine the iron oxidation state.

**Tetrairon(III) Orthocarbonate $Fe_4^{3+}C_3O_{12}$.** We observed appearance of the new sharp spots in the diffraction pattern after laser heating of $FeCO_3$ single crystals at 1,750(100) K at 74(1) GPa (Fig. 1, Supplementary Table 1). Single-crystal X-ray diffraction (for example, Supplementary Fig. 1) data were collected on a temperature-quenched sample (Supplementary Tables 2 and 3). Reflections of one of the phases were indexed in a

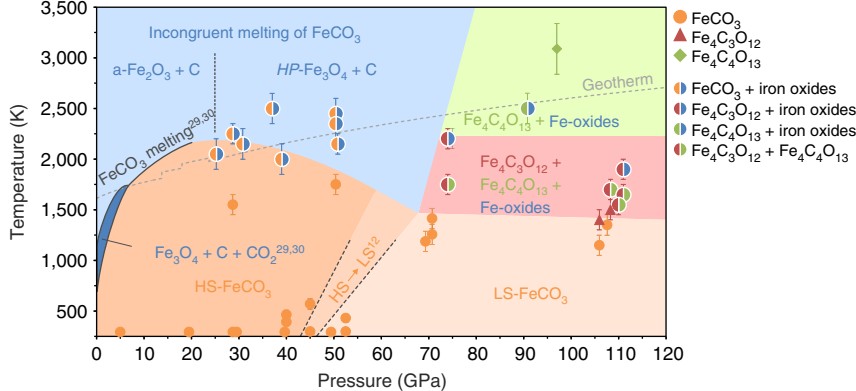

**Figure 1 | Stability diagram of FeCO₃ at high P-T.** Symbols and phase regions identified in experiments: magnesite-structured $FeCO_3$ (orange dots), oxide(s) and recrystallized $FeCO_3$ (orange-blue dots), tetrairon(III) orthocarbonate $Fe_4C_3O_{12}$ (red triangles), diiron(II) diiron(III) tetracarbonate $Fe_4C_4O_{13} + Fe_4C_3O_{12}$ (red-green dots), $Fe_4C_4O_{12} +$ oxides (red-blue dots), $Fe_4C_4O_{13} +$ oxides (green-blue dots), $FeCO_3$ decomposition to $Fe_3O_4 + C + CO_2$ (refs 29,30; dark blue area), high- and low- spin $FeCO_3$ (dark and light orange area, respectively), incongruent melting of $FeCO_3$ (light blue area), and formation of HP-carbonates $Fe_4C_3O_{12}$ and $Fe_4C_4O_{13}$ (red and green areas). The grey dashed curve is the expected mantle geotherm[59]. The black solid lines are from refs 29,30. Black dashed lines indicate the region delimiting the spin transition in magnesio-siderite at HPHT from Liu et al.[13]. The vertical dashed black line separates the regions in which the formation of α-$Fe_2O_3$ and HP-$Fe_3O_4$ was observed upon incongruent melting of $FeCO_3$.

hexagonal unit cell (space group R3c, #161). After the integration procedure the resulting data set contained 327 reflections with $I > 2\sigma$ (I). Structure solution using direct methods identified the phase as a novel iron carbonate with stoichiometry $Fe_4C_3O_{12}$. Charge balance considerations show that all iron is ferric. Each carbon atom is surrounded by four oxygen atoms (C–O distances $\sim 1.31–1.40$ Å at 74 GPa), forming isolated (that is, not linked to each other) tetrahedra (Fig. 2a,b). Thus, the new compound is referred to as tetrairon(III) orthocarbonate.

There are two structurally distinct iron positions in the phase: the Fe(1) atom is situated on a threefold axis, while Fe(2) occupies a general position (Supplementary Table 2). Fe(1) is surrounded by nine oxygen atoms where three of the oxygen atoms are much further away than the other six ($\sim 2.30$ Å versus $\sim 1.94$ and 1.97 Å at 74(1) GPa); hence this polyhedron can be considered to be a regular trigonal prism (Fig. 2a). The Fe(2) atom is located in a bicapped trigonal prism coordinated by oxygen. The individual (and average) Fe–O distances in both of the iron polyhedra in $Fe_4C_3O_{12}$ are longer than in low-spin iron(III)-bearing oxides and compounds ($\sim 1.8$ Å) at corresponding pressures, which suggests that the phase contains high-spin iron. Although a precise characterization of the pure phase using Mössbauer spectroscopy is difficult due to the presence of other iron compounds (particularly iron oxides, see below) in the laser-heated samples, the available information supports the presence of iron in the high-spin state (see Supplementary Fig. 2, Supplementary Table 4 and Supplementary Note 1).

The threefold symmetry ring formed by corner- and edge-shared $CO_4^{-}$ tetrahedra and three Fe(2)O8-bicapped prisms is a notable characteristic of the tetrairon orthocarbonate structure (Fig. 2a). The rings form layers that are stacked along the c axis (where each subsequent layer is rotated by 120° with respect to the original one, Fig. 2a). The trigonal Fe(1)O6-prisms are connected via triangular bases that are located in the channels formed by stacked rings. We are not aware of any other compounds that form the same structure.

**Diiron(II) Diiron(III) Tetracarbonate $Fe_2^{2+}Fe_2^{3+}C_4O_{13}$.** Laser heating of $FeCO_3$ at temperatures above 1,750(100) K at pressures above 74(1) GPa resulted in formation of not only $Fe_4C_3O_{12}$ and iron oxides (see below), but also a monoclinic phase (space group C2/c, #15; Supplementary Tables 1 and 2). Single-crystal X-ray diffraction data were collected on temperature-quenched samples at different pressures, where the best results were obtained for the experiment at 97(2) GPa (Fig. 1, Supplementary Tables 1 and 2). The chemical composition derived from the structure solution is $Fe_4C_4O_{13}$, or more specifically $Fe^{2+}_2Fe^{3+}_2C_4O_{13}$. Each carbon atom is tetrahedrally coordinated by four oxygen atoms (C–O distances $\sim 1.27–1.39$ Å at 97(2) GPa), and four $CO_4$ groups are linked in truncated chains (Fig. 2b, Supplementary Fig. 4b). Thus, we refer to the new compound as diiron(II) diiron(III) tetracarbonate.

The atomic arrangement of the structure is based on corner-linked infinite layers of Fe(2)O8-bicapped prisms connected in a three-dimensional (3D) framework by dimers of edge-shared Fe(1)O7 monocapped prisms and zigzag-shaped $C_4O_{13}^{10-}$ chains (Fig. 2b). The average Fe–O distances in Fe(1)O7 and Fe(2)O8

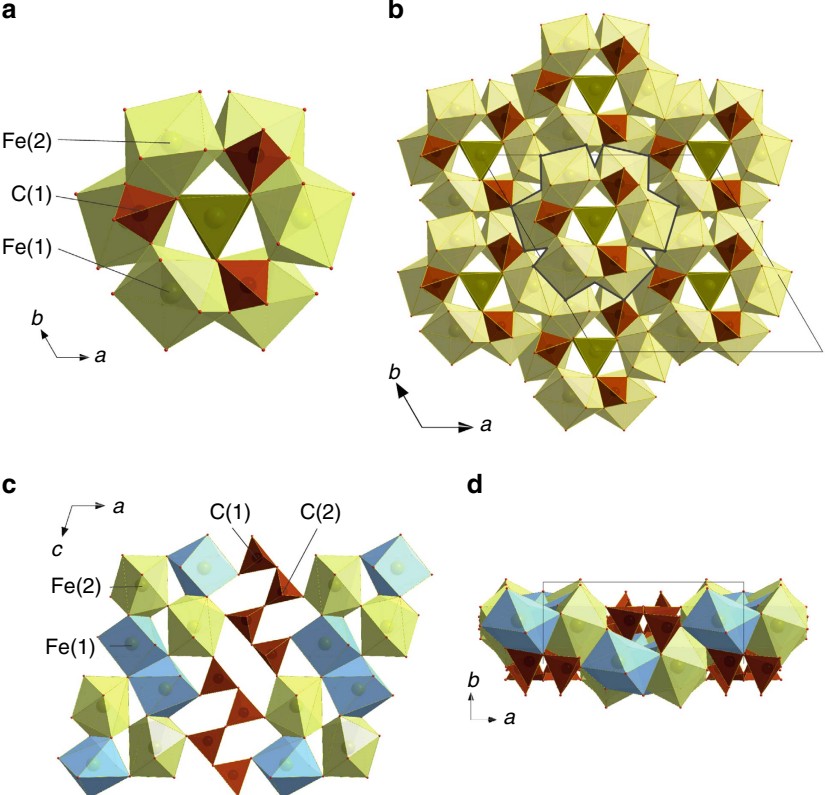

**Figure 2 | Crystal structures of high-pressure carbonates.** (**a,b**) Tetrairon(III) orthocarbonate $Fe_4C_3O_{12}$ and (**c,d**) diiron(II) diiron(III) tetracarbonate $Fe_4C_4O_{13}$, at ambient temperature and 74(1) and 97(2) GPa, respectively. In **a**, three FeO8 bicapped prisms (light green) and three $CO_4^{-}$ tetrahedra (brown) form a ring with threefold symmetry by corner and edge sharing. The rings form layers that are stacked along the c axis. FeO6-prisms (dark green) are connected by triangular bases and located in the channels created by the rings. In **b** the overall structure of the orthocarbonate is displayed along the c-axis. In **c** FeO8 bicapped prisms (light green) are connected in a 3D framework by dimers of edge-shared FeO7 monocapped prisms (blue) and zigzag-shaped $C_4O_{13}^{10-}$ chains (brown) shown along the b axis. In **d** the tetracarbonate structure is displayed along the c axis.

polyhedra are similar to one another ($\sim$1.96 Å and $\sim$2.02 Å, respectively, at 97(2) GPa) and longer than expected for low-spin ferric or ferrous iron[26]. Similarity in the sizes of iron polyhedra may indicate that Fe cations are in a mixed valence state (intermediate between $+2$ and $+3$) as proposed for high-pressure iron oxides[26,27].

$Fe_4C_4O_{13}$ is isostructural with recently reported $Mg_{1.6}Fe_{2.4}C_4O_{13}$ (ref. 24) obtained by annealing Mg-bearing natural siderite at 141 GPa and 2,650 K. Indeed there is an entire family of tetrasilicates containing four-member $Si_4O_{13}^{10-}$ groups (ref. 28 and references therein), as well as germanates, vanadates and arsenates.

**FeCO₃ behaviour at high pressures and high temperatures.** Recent studies of $FeCO_3$ at pressures up to 20 GPa (refs 29,30) have shown that while it melts above $\sim$6 GPa, below this pressure the phase dissociates to the mixture $Fe_3O_4 + C + CO_2$ before melting. Heating $FeCO_3$ single crystals at 28.7(1) GPa and 1,550(100) K does not result in any phase or structural changes or destruction of the single crystal (Fig. 3b). Heating the same sample at higher temperatures (2,250(100) K) causes complete recrystallization of the material (Fig. 3c), thus indicating melting. Mössbauer spectra of the samples quenched from the molten state show unambiguously the presence of $FeCO_3$ as well as iron oxides: $\alpha$-$Fe_2O_3$ (hematite) at pressures below $\sim$25 GPa and HP-$Fe_3O_4$ (ref. 26) above $\sim$31 GPa (Figs 1 and 4b,c,f,g; Supplementary Tables 1,4 and 5).

We observed a similar behaviour for $FeCO_3$ upon heating to $\sim$51 GPa (Fig. 1, Supplementary Tables 1 and 5). However, while the degree of carbonate decomposition appears to increase at increasing temperature (at a given pressure) and increasing pressure (at a given temperature), we cannot quantify the process based on our existing data. Nevertheless it is clear that all experiments performed below $\sim$51 GPa produced only partial decomposition of the carbonate. Even heating above $\sim$2,450 K for up to one hour always showed diffraction lines of recrystallized $FeCO_3$ and/or its presence in SMS spectra. However, we cannot exclude that heating at sufficiently high temperature (and for a sufficiently long time) could result in complete breakdown of $FeCO_3$.

$Fe^{2+}$ in $FeCO_3$ is known[17–21] to undergo spin crossover from high to low spin at about 40 GPa. Although the goal of this work was not to investigate the pressure–temperature dependence of spin crossover, Mössbauer spectroscopy (SMS) was able to show the temperature effect on the spin state at different pressures (examples of spectra are given in Fig. 4). The corresponding data

points are shown on Fig. 1 and are reasonably consistent with the data reported by Lin et al.[21] and Liu et al.[22] for magnesio-siderite $Mg_{0.35}Fe_{0.65}CO_3$.

Heating $FeCO_3$ at 74(1) GPa and 1,750(100) K resulted in the formation of multi-domains of both tetrairon orthocarbonate $Fe_4C_3O_{12}$ and diiron(II) diiron(III) tetracarbonate, $Fe_4C_4O_{13}$

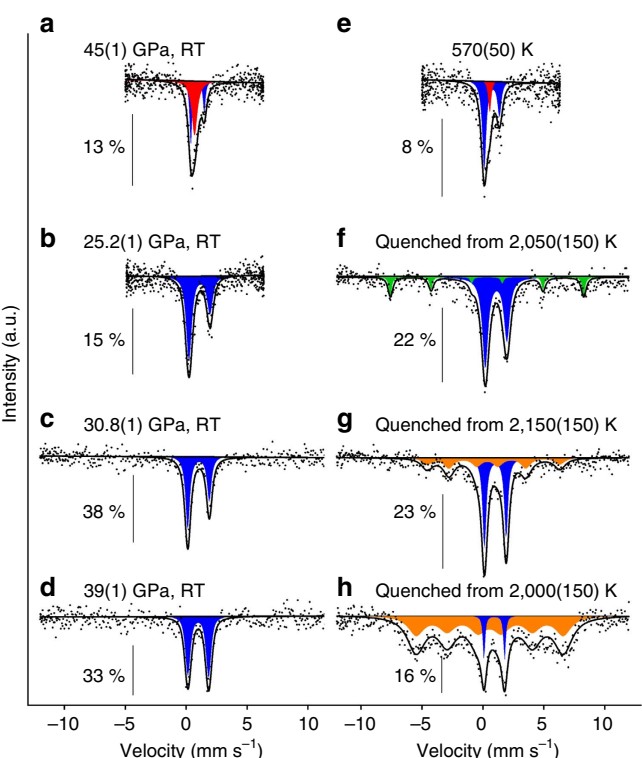

**Figure 4 | Selected synchrotron Mössbauer spectra from high *P-T* treatment of FeCO₃.** (**a–d**) Spectra collected at the indicated pressures at room temperature before heating; (**e–h**) Spectra collected during or after heating at the indicated temperatures. Subspectra are shaded as follows: blue doublet—high-spin ferrous iron, red singlet—low-spin ferrous iron; green sextet—$\alpha$-$Fe_2O_3$ and orange sextet—HP-$Fe_3O_4$ (Supplementary Fig. 5). Note that in **a** the intensity (amount) of high-spin ferrous iron increases with increasing temperature. The appearance of HP-$Fe_3O_4$ in **g** and **h** is the result of progressive decomposition of $FeCO_3$ upon melting at high pressure. Continuous lines are fitted with the full transmission integral (Methods section). Percentages on the left of each spectrum indicate the relative absorption.

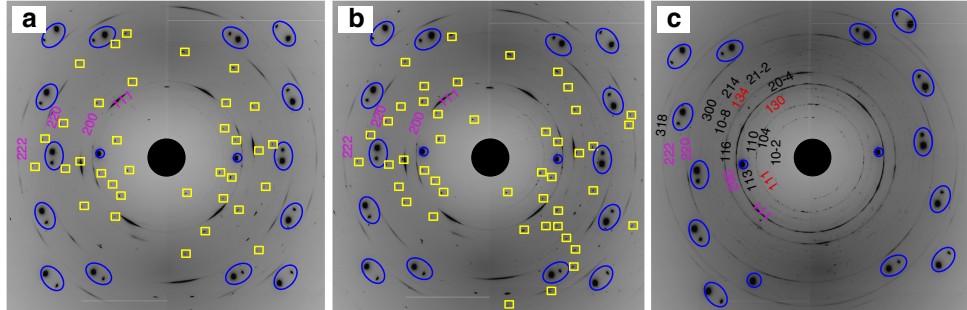

**Figure 3 | Selected 2D diffraction images of FeCO₃ before and after heating.** Single-crystal diffraction images were collected at 28.7(1) GPa at (**a**) ambient temperature before heating, (**b**) after heating at 1,550(100) K and (**c**) 2,250(100) K. Melting and recrystallization of starting material is evidenced by the appearance of $FeCO_3$ powder diffraction rings after heating to the highest temperature. Appearance of diffraction lines of HP-$Fe_3O_4$ indicates incongruent melting of $FeCO_3$. Blue ellipsoids indicate diamond diffraction spots from the diamond anvils and yellow squares mark the single-crystal diffraction spots of $FeCO_3$. Numbers designate hkl-indices of diffraction lines belonging to $FeCO_3$ (black), neon pressure transmitting medium (magenta) and HP-$Fe_3O_4$ (red).

(Fig. 1). The result is reproducible: the simultaneous appearance of both phases was observed at the same pressure and temperature on heating of different crystals in different loadings (Supplementary Table 1). The observations may (a) indicate non-equilibrium conditions in laser-heated DACs, (b) be a consequence of complex redox chemical reactions that form both phases simultaneously or (c) be a result of successive transformations between carbonate phases that are so closely located in P-T space that they cannot be distinguished. Indeed the results of further experiments at higher pressures (103–110 GPa) suggests that scenario (c) is the most plausible. Heating a $FeCO_3$ single crystal to 1,400(100) K at 110(2) GPa resulted in the formation of hexagonal tetrairon orthocarbonate $Fe_4C_3O_{12}$ (Figs 1 and 5). Monoclinic diiron(II) diiron(III) tetracarbonate, $Fe_4C_4O_{13}$ appeared only after laser heating above 1,650(100) K

(Figs 1 and 5). The highest temperature at which $Fe_4C_4O_{13}$ was observed is 3,088(250) K at 97(2) GPa (Fig. 1).

Close examination of X-ray diffraction patterns containing diiron(II) diiron(III) tetracarbonate, $Fe_4C_4O_{13}$ reveal the presence of further reflections which do not belong in any obvious way to the previously identified carbonates. While in some cases it was not possible to identify phase(s) unambiguously due to close overlapping of reflections and/or their low amount, for most data points we were able to determine lattice parameters and even refine structure(s). In all cases, the appearance of $Fe_4C_4O_{13}$ upon heating at pressures above ~74 GPa was associated with the formation of orthorhombic (space group Cmcm) CaIrO$_3$-structured $\eta$-Fe$_2$O$_3$ (ref. 26) and/or orthorhombic (space group Bbmm) CaTi$_2$O$_4$-type structured HP-Fe$_3$O$_4$ phase[26,31] (Fig. 1, Supplementary Tables 1 and 5). Additionally, in several experiments after heating at different pressures but at temperatures above ~2,200 K, we observed monoclinic (space group C2/m) Fe$_5$O$_7$ (ref. 26).

Samples recovered after experiments at pressures above ~30 GPa at temperatures higher than ~1,800 K show strong broad features in Raman spectra that are typical for nano-diamond[32] and sometimes a band at 1,330 cm$^{-1}$ that is characteristic for diamond (Supplementary Fig. 3), even though in situ X-ray measurements were not able to detect the presence of diamond (or any other carbon phases) after laser heating of $FeCO_3$ crystals. The formation of diamond upon heating of $FeCO_3$ at high pressures and temperatures has been unambiguously documented in previous studies[12,29,30].

## Discussion

The increase of carbon co-ordination number from three (CO$_3$ triangles) to four (CO$_4$ tetrahedra) under compression is the obvious consequence of rules known for decades[33]. Numerous theoretical studies have predicted the formation of CO$_4$-bearing carbonates at pressures ranging from over 80 GPa to 150 GPa[34–37] depending on chemical composition and computational methods. Different configurations, from isolated tetrahedra to pyroxene-like chains, were anticipated for compounds with carbon tetra-co-ordinated by oxygen. In general, theoretical analysis of possible carbonates with condensed CO$_4$ groups suggests that there should be analogues with silicates, but expected variations of C–O–C angles are much smaller than for Si–O–Si angles in silicates[36,38].

We are not aware of dedicated theoretical studies of pure iron CO$_4$-bearing carbonates. However, the predicted[36] structure of Mg-carbonate containing three-membered rings C$_3$O$_9^{6-}$ made of CO$_4$ tetrahedra was used to index peaks in powder diffraction data of HPHT Fe- and Fe/Mg carbonates[12,16]. The same structural model has been used to interpret infrared spectra of magnesiosiderite[39]. However, important structural details predicted by theory[36] and obtained by powder X-ray diffraction experiments[12,16] are not in agreement (even reported space groups are different). Moreover, it is not obvious how reliable structural information can be extracted based on LeBail fits with large unit cells[13] or from powder X-ray diffraction of complex mixtures of different phases: for example, α-Fe$_2$O$_3$ (hematite) was one of the phases reported to co-exist with carbonates at 88 GPa (ref. 12) while it is well documented by now[26] that above ~50 GPa iron(III) oxide adopts different structures.

The average interatomic C–O distance in the CO$_3^{2-}$ ion in magnesite-structured $FeCO_3$, MnCO$_3$, and CoCO$_3$ (refs 17,40) extrapolated to 75 GPa is ~1.24 Å, and is ~1.26 Å[9] in dolomite-III. In iron ortho- and tetracarbonates the average C–O distance is ~1.35 Å, where the larger value is consistent with the increase of coordination number on transformation and quantitatively similar to borates (the typical difference in ionic radius of B$^{3+}$ in

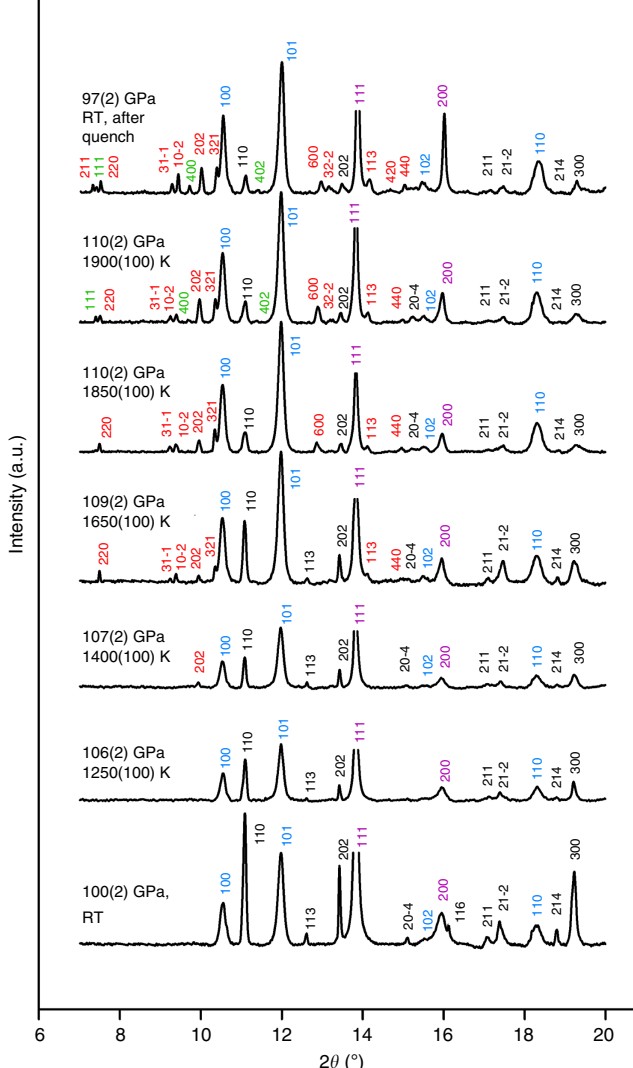

**Figure 5 | Selected integrated X-ray diffraction patterns collected on heating of FeCO$_3$ near or above 100 GPa.** Patterns are labelled with the pressure and temperature during data collection. Indices of different phases are coloured as follows: FeCO$_3$ starting material (black), Ne (magenta), Re (light blue), tetrairon(III) orthocarbonate Fe$_4$C$_3$O$_{12}$ (red) and diiron(II) diiron(III) tetracarbonate Fe$_4$C$_4$O$_{13}$ (green). Neon was used both as a quasi-hydrostatic pressure medium and a pressure standard (Methods section). Data were collected at ID09a at ESRF with an X-ray wavelength of 0.4126 Å.

triangular $BO_3$ and $BO_4$ tetrahedra is $\sim 0.1$ Å)[33]. Notably, the average O–O distance (2.20(2) Å) in the $CO_4^{4-}$ group in iron orthocarbonate at $\sim 74$ GPa is, for example, almost the same as in magnesite-structured Fe, Mn and Co carbonates ($\sim 2.15$ Å) and in dolomite-III ($\sim 2.17$ Å) (refs 17,40). This observation indicates that is not the size of the carbon cation, but rather the oxygen–oxygen contacts that define the size of $CO_4$ tetrahedra.

The shapes of $CO_4$ tetrahedra in HP-carbonates are far from ideal. The polyhedron is especially distorted in tetrairon orthocarbonate $Fe_4C_3O_{12}$ with dissimilar individual C–O distances (varying from 1.254 to 1.385 Å at 103.7(2) GPa) and O–C–O angles (varying from 98 to 115° with bond angle variance of 61.4 degree$^2$, see Supplementary Note 2 for definition and reference). While orthosilicates display a great variety of geometries, such distortion of $SiO_4$ units is not unusual and garnets in particular show bond angle variances in the same range (57 degree$^2$ for pyrope, for example, ref. 41).

In tetracarbonate there are two non-equivalent $CO_4$-tetrahedra, two 'outer' and two 'inner' that form four-membered $C_4O_{13}^{10-}$ truncated chains (Fig. 2c,d). In both Mg-bearing[24] and pure iron tetracarbonate the 'inner' tetrahedra are significantly more distorted than the 'outer': the bond angle variances are 35.9 degree$^2$ compared to 13.1 degree$^2$ for pure iron tetracarbonate at 97(2) GPa, and 146.2 compared to 19.0 degree$^2$ for the Mg-bearing phase at 135 GPa (ref. 24). Indeed the same tendency is observed in tetrasilicates (ref. 28 and references therein). Moreover, the Si–O–Si angles in tetrasilicates are $\sim 122°$ and 125–143° (for 'outer'-'inner' and 'inner'-'inner' tetrahedra, respectively[28]), while for tetracarbonates the corresponding C–O–C angles are $\sim 113°$ and 122–125°. Thus, the analogous structural behaviour of silicates and $CO_4$-based carbonates is obvious: based on experimental observations there is no reason that 'high-pressure' carbonate structures should be more limited than those of silicates. Moreover, iron(III) orthocarbonate, $Fe_4C_3O_{12}$, with its unique structure is an example of the diversity of atomic arrangements that are possible in high-pressure $CO_4$-bearing carbonates.

Heating $FeCO_3$ above 1,750(100) K at pressures to at least $\sim 50$ GPa (Fig. 1) resulted in partial decomposition of the material with formation of iron oxides ($\alpha$-$Fe_2O_3$ below $\sim 25$ GPa and high-pressure orthorhombic HP-$Fe_3O_4$ at higher pressure). These observations are in agreement with results obtained in large volume apparatus: for example, according to Kang et al.[30], $FeCO_3$ partially decomposes on melting according to the following reaction:

$$6\,FeCO_3 = 2\,Fe_3O_4 + C + 5\,CO_2 \qquad (1)$$

Formation of tetrairon(III) orthocarbonate, $Fe_4C_3O_{12}$, at pressures above $\sim 74$ GPa (Fig. 1) may be described by the simple equation:

$$4\,FeCO_3 = Fe_4C_3O_{12} + C \qquad (2)$$

According to equation (2) and known equations of state of $FeCO_3$ (with low-spin $Fe^{2+}$) and diamond[17,42] the gain in volume on decomposition of $FeCO_3$ is $\sim 4\%$. Thus, redox reaction (2) may be driven by a volumetric effect.

Diiron(II) diiron(III) tetracarbonate, $Fe_4C_4O_{13}$, appears on heating of $FeCO_3$ above $\sim 74$ GPa, but at temperatures significantly higher than needed to form tetrairon(III) orthocarbonate (Fig. 1). The experiment at $\sim 110$ (Fig. 5) GPa indeed demonstrates that tetracarbonate forms after further heating of already synthesized orthocarbonate. Thus, we can conclude that tetracarbonate is a product of the chemical evolution of orthocarbonate. Some schemes that may lead to formation of

tetracarbonate are the following:

$$13\,Fe_4C_3O_{12} + 9\,C = 12\,Fe_4C_4O_{13} + 4\,Fe, \qquad (3)$$

or

$$7\,Fe_4C_3O_{12} + 3\,C = 6\,Fe_4C_4O_{13} + 2\,Fe_2O_3. \qquad (4)$$

Equations (3) and (4) suggest that carbon (as a decomposition product of $FeCO_3$ according to equation (2) or as diamond from the anvils) reduces $Fe^{3+}$. However, this process is unlikely because carbon (diamond) cannot reduce iron at pressures above $\sim 25$ GPa (refs 15,26). Moreover, we did not observe evidence of pure iron or its alloys (or carbides) in reaction products, which adds to the arguments against reaction (3).

Another process which could lead to formation of tetrairon tetracarbonate is

$$8\,Fe_4C_3O_{12} = 6\,Fe_4C_4O_{13} + 4\,Fe_2O_3 + 3\,O_2. \qquad (5)$$

In this case iron is reduced by oxygen. The same phenomenon was observed in studies of iron(III) oxide at pressures above $\sim 70$ GPa (ref. 26). Moreover, together with iron tetracarbonate we observed (Fig. 1, Supplementary Tables 1 and 5) $CaIrO_3$-structured $\eta$-$Fe_2O_3$ and possible products of its further decomposition[24,26]: monoclinic (space group C2/m) $Fe_5O_7$, monoclinic (space group C2/m) $Fe_{13}O_{19}$ and orthorhombic (space group Bbmm) $CaTi_2O_4$-type structured HP-$Fe_3O_4$ phases. These observations support the schematic process described by equation (5) and imply that the presence of iron oxides among products of HPHT treatment of iron carbonate(s) above 74(1) GPa is not a signature of their full breakdown, but rather an indication of the intrinsic process of formation of $CO_4$-bearing phase(s) containing iron in different oxidation states.

While only reaction (5) explicitly depends on oxygen fugacity, all other processes described above may be affected by different redox conditions (and could play a role in buffering reactions in more complex processes involving iron-bearing carbonates). In view of the recently reported[26,43] fundamental changes in the chemical behaviour of the iron-oxygen system at pressures above $\sim 70$ GPa and high temperatures, our results are also calling for detail investigations of redox processes in the lowermost part of Earth's lower mantle and core–mantle boundary.

Heating magnesio-siderite $(Fe_{0.65}Mg_{0.35})CO_3$ at pressures of $\sim 50(1)$ GPa and 1400(100) K resulted[13] in formation of a new phase which Liu et al.[13] called 'siderite II' and described as orthorhombic. The same phase was reported[13] at pressures up to $\sim 120$ GPa and temperatures up to 2,200 K. Liu et al.[13] provided us with the powder X-ray diffraction pattern collected at 90 GPa after heating at 2,200 K. We fit this data using the Rietveld method (as implemented in the GSAS package) using the model of hexagonal tetrairon orthocarbonate $Fe_4C_3O_{12}$ (Supplementary Fig. 4). The good quality of the fit (Supplementary Fig. 4) unambiguously confirms that 'siderite II' has the structure of tetrairon orthocarbonate. The quality of the powder X-ray diffraction data does not allow the occupancy of cation positions to be refined so we do not know how much magnesium is incorporated in the phase. However, the absence of reflections of any other phases (apart from the gold standard and Ne pressure transmitting medium) suggests that at least 35% of magnesium may be incorporated in the structure of orthocarbonate.

Single-crystal X-ray diffraction data on iron-bearing carbonates subjected to high pressures and high temperatures are very limited. Studies of ankerite[9] up to about 60 GPa reveal the formation of a phase with non-planar $CO_3^{2-}$ groups as a tendency to increase the co-ordination number of carbon. Magnesium-siderite exposed to pressures and temperatures corresponding to the top of the Earth's D″ layer (135 GPa and

2,650 K) was shown[24] to transform to iron(II)-bearing dimagnesium diiron(III) tetracarbonate $Mg_{1.6}Fe_{2.4}C_4O_{13}$. It contains tetrahedrally coordinated carbon units, corner-shared in truncated $C_4O_{13}^{10-}$ chains, and up until the present work it was the only unambiguously proven case of carbonate with $CO_4$ groups.

In the case of tetrairon orthocarbonate, incorporation of magnesium may significantly expand its pressure–temperature stability field at the expense of the diiron diiron tetracarbonate phase. It may mean that Mg, Fe-orthocarbonate might remain stable along the geotherm.

Modern estimates indicate the concentration of carbon in altered oceanic crust to be in the range of 500–600 p.p.m. (ref. 5 and references therein). A recent study concludes that between none and almost all of the carbonates in subducting slabs are subducted into the mantle[5]. The most plausible scenario however is that a relatively small amount of carbon is recycled into the convecting mantle based on a careful re-evaluation of carbon fluxes in subduction zones, where carbon is preferentially stored in the lithospheric mantle and the crust[5]. Our experiments clearly show that along an average subducting slab temperature profile[44] (Fig. 1), Fe-carbonates will not melt during subduction. Still, recent reports[4] suggest that the majority of slab geotherms intersect a deep depression along the melting curve of carbonated oceanic crust at depths of $\sim 300$–700 kilometres, creating a barrier to direct carbonate recycling into the deep mantle. Nevertheless, a portion of the subducted carbonates can still be recycled in the convecting mantle, not taking part in hydrothermal alteration reactions, partial melting and formation of carbonatitic magmas and 'premature' carbon degassing processes. For instance, cold subducting slabs[44,45] (Fig. 1) could stabilize carbonates down to mid lower mantle depths, favouring Fe-partitioning into carbonates[13] due to $Fe^{2+}$ high- to low-spin crossover, which starts in carbonates at much shallower depths than in other $Fe^{2+}$-bearing minerals[46,47]. In this case, the average composition of carbonates in the Earth's lower mantle could be significantly enriched towards the '$FeCO_3$' component. While self-redox reactions involving iron may potentially destabilise Fe-bearing carbonates, our experiments demonstrate that at pressures above $\sim 70$ GPa (corresponding to a depth of $\sim 2,000$ km), iron carbonates drastically change their structures, forming $CO_4$-bearing compounds, and may persist to temperatures above 3,000 K in the 100 GPa range (that is, exist above the mantle geotherm). Thus, based on our experimental observations we conclude that $CO_4$-based carbonates may be carriers of carbon in the lower mantle.

## Methods

**Sample preparation.** Single crystals of $^{57}FeCO_3$ were grown from $^{57}FeCO_3$ powder at 18 GPa and 1,600 °C in a 1,200-t Sumitomo press at Bayerisches Geoinstitut (Bayreuth, Germany). $^{57}FeCO_3$ powder was synthesized using $^{57}Fe$-oxalate ($^{57}FeC_2O_4$) as a precursor, which in turn was obtained via chemical reactions starting from $^{57}Fe$-metal (see ref. 20 for more details). Mössbauer spectroscopy and single-crystal diffraction data confirm that synthetic samples are free of ferric iron.

Single crystals with an average size of $0.015 \times 0.015 \times 0.005$ mm$^3$ were pre-selected on a three-circle Bruker diffractometer equipped with a SMART APEX CCD detector and a high-brilliance Rigaku rotating anode (Rotor Flex FR-D, Mo-K$\alpha$ radiation) with Osmic focusing X-ray optics.

Selected crystals together with small ruby chips (for pressure estimation) were loaded into BX90-type DACs[48] and European Synchrotron Radiation Facility (ESRF) high-pressure membrane cells. Diamonds with culet sizes of 250 μm and 120 μm in diameter were used to generate pressures up to $\sim 75$ GPa and $\sim 110$ GPa, respectively. Neon was used as a pressure transmitting medium and was loaded at Bayerisches Geoinstitut and/or at ESRF.

**X-ray diffraction.** The single-crystal X-ray diffraction experiments were conducted on the ID09a beamline at ESRF, Grenoble, France (MAR555 detector, $\lambda = 0.4126$ Å); on the ID27 beamline at ESRF (PerkinElmer flat panel detector,

$\lambda = 0.3738$ Å); and on the 13-IDD beamline at the advanced photon source (APS), Chicago, USA (MAR165 CCD detector, $\lambda = 0.3344$ Å). The X-ray spot size was dependent on beamline settings and varied from, for example, $3 \times 2.5$ μm$^2$ (13-IDD) to $10 \times 10$ μm$^2$ (ID09a), where typically a smaller beam was used for laser heating experiments. A portable double-sided laser heating system[49] was used for experiments on ID09a (ESRF) to collect *in situ* single-crystal X-ray diffraction data, while a state-of-the art stationary double-sided laser heating setup at IDD-13 (APS) was used to collect temperature-quenched single-crystal X-ray diffraction data. Crystals (as a rule about 10 μm in diameter) were completely immersed in laser radiation and there was no measurable temperature gradient within the samples. In the case of prolonged heating experiments the temperature variation during the heating did not exceed $\pm 100$ K. Pressures were calculated from the positions of the X-ray diffraction lines of Ne (ref. 50). X-ray diffraction images were collected during continuous rotation of DACs typically from $-38°$ to $+38°$ on $\omega$; while data collection experiments were performed by narrow 0.5–1° scanning of the same $\omega$ range.

Integrated patterns (Fig. 5) from powder XRD experiments were processed using Fit2d (ref. 51) and indexed using the Rietveld method implemented in the GSAS and EXPUI packages[52,53].

**XRD data analysis.** Integration of the reflection intensities and absorption corrections were performed using CrysAlisPro software[54]. Diffraction images were converted to the native CrysAlisPro format 'ESPERANTO'. An instrument model was refined to calibrate the diffractometer. For this purpose we performed data collection on standard samples. Calibration of the instrument model for Fit2D software[51] was done using powder standards, that is, LaB$_6$ (NIST SRM 660a) or CeO$_2$ (NIST SRM 674b) and for CrysAlisPro software we used a standard orthoenstatite calibration single crystal ((Mg$_{1.93}$,Fe$_{0.06}$)(Si$_{1.93}$,Al$_{0.06}$)O$_6$, $Pbca$, $a = 8.8117(2)$, $b = 5.18320(10)$, $c = 18.2391(3)$ Å, already mounted in a DAC). The peak hunting of the experimental data set was performed using the smart peak hunting option in CrysAlisPro for images collected with MAR165 (13 = IDD beamline at APS and ID27 beamline at ESRF) and Perkin Elmer (ID27 beamline at ESRF) detectors. For images collected with a MAR555 detector (ID09a) we utilized the traditional peak hunting procedure. To obtain the unit cell parameters of the measured phases(s) we proceeded with the automatic indexing of the peaks. The best performance of the automatic indexing method can be achieved on a small set of reflections ($\sim 20$–30) belonging only to a single crystal. One has to manually select those reflections that build a 3D lattice in the reciprocal space. The obtained unit cell was then refined against the whole batch of reflections and the UB matrix was derived. Once the unit cell parameters were defined, we proceeded with the data reduction that is necessary to extract the reflection intensities from the experimental images. By default, after data reduction, CrysAlisPro applies frame scaling, absorption corrections and searches for the space group analysing the systematic absences. This operation requires the following integration parameters: correct data ranges, DAC's opening angle, integration box size, reflection profile fitting mode (2D or 3D) and background evaluation mode. If the reflection profiles were split over several frames, the 3D profile fitting was used, the 2D option otherwise. Data from laser-heated samples (noisy data) required the smart background option to be used. Data reduction output files (including intensity and resolution statistics) were inspected manually to check for consistency between intensities of equivalent reflections. Indicators of the XRD data quality are $R\sigma$, $F^2_{obs}/\sigma_{int}(F^2_{obs})$ and $R_{int}$ and a form of frame-by-frame scaling coefficients (frame scaling curve). The $R_{int}$ value indicates the overall quality of the data collection and if it is higher than 10% an accurate structural refinement will not be possible. Processes of elaboration of phases are nowadays automated, and there are a number of techniques implemented in different structure solution programs such as direct methods, Patterson synthesis, heavy-atom method, charge flipping and so on. Here, the structures were solved by the direct method and refined in the isotropic (and anisotropic for iron atoms) approximation by full matrix least-squares refinements using SHELXS and SHELXL software[55], respectively. The atomic co-ordinates were calculated using an inverse Fourier transform for the structure factor $F_{hkl}$ of the specific (for each phase) diffracted waves. Once we obtained the initial structural model we refined it against the experimental data by least-squares minimization of adjustable parameters. At the first stage, missing atoms were found from the reconstruction of residual electron density maps, their position, and when applicable, atomic occupancies were refined.

High-pressure data normally suffer from overlapping with parasite diffractions, mostly created by diamonds and crystallized pressure media. Those overlapping reflection were omitted from the refinement.

Note that all information given here can be found illustrated in https://epub.uni-bayreuth.de/2124/.

**SMS spectroscopy.** Energy-domain Mössbauer spectroscopy measurements were carried out at the nuclear resonance beamline ID18 (ref. 56) at ESRF using the SMS. The SMS is based on a nuclear resonant monochromator employing pure nuclear reflections of an iron borate ($^{57}FeBO_3$) crystal. The source provides $^{57}Fe$ resonant radiation at 14.4 keV within a bandwidth of 15 neV which is tunable in energy over a range of $\sim \pm 0.6$ μeV (ref. 56). The beam of $\gamma$-radiation emitted by the SMS was focused to a 10 μm $\times$ 15 μm spot size, fully within the size of the Fe-carbonate crystals. During each laser heating experiment we measured the Mössbauer

spectrum of the sample before and after heating. We did not measure the Mössbauer spectrum of the sample during heating with exception of the experiments performed in the spin crossover pressure range, in between ~40 and ~50 GPa (Fig. 1 and Supplementary Table 1). The small cross section of the beam and its high intensity allow for rapid collection of Mössbauer data[57]. The collection time for one $FeCO_3$ spectrum before heating was ~20 min, whereas after heating (depending on the sample composition) the collection time varied from 30 min to 6 h (the latter for the new HP-carbonates). The velocity scales of all Mössbauer spectra were calibrated relative to 25 μm-thick α-Fe foil, and all spectra were fitted using the software package MossA (ref. 58). Lorentzian lines were used to fit all Mössbauer spectra and a linear function was applied to model the background. All spectra were fit using the full transmission integral to avoid distortion of the area ratios due to the high loading of $^{57}Fe$ in our samples. In this way we were able to obtain an accurate determination of iron distribution in the different investigated samples such as the high spin and low spin abundance and site population in HP-carbonates.

**Data availability.** The X-ray crystallographic co-ordinates for structures reported in this article have been deposited at the Inorganic Crystal Structure Database (ICSD) under deposition number CSD 432930-432931. These data can be obtained free of charge from FIZ Karlsruhe, 76344 Eggenstein-Leopoldshafen, Germany (fax: (+49)7247-808-666; e-mail: crysdata@fiz-karlsruhe.de) through the hyperlink: https://www.fiz-karlsruhe.de/en/leistungen/kristallographie/kristall-strukturdepot/order-form-request-for-deposited-data.html

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

## Acknowledgements

We thank G. Aprilis, D. Vasiukov, I. Chuvashova, A. Pakhomova and N. Solopova for assistance with HPHT experiments at ESRF. We acknowledge the helpful collaboration of J. Liu & J.-F. Lin and the technical assistance of S. Linhardt and S. Übelhack. L.D. and C.M. thank the German Research Foundation (Deutsche Forschungsgemeinschaft, DFG) and the Federal Ministry of Education and Research (BMBF, Germany) for funding. M.B. thanks the Ministry of Education and Science of the Russian Federation in the framework of Increase Competitiveness Program of NUST "MISIS" (No. K2-2016-013). We acknowledge the ESRF for provision of beam time at ID09a, ID18, ID27 and the Sample Environment Service-HP lab for the technical support of their loan pool diamond anvil cells. Portions of this work were performed at Geo-SoilEnviroCARS (sector 13), APS, Argonne National Laboratory. GeoSoilEnviroCARS is supported by the National Science Foundation-Earth Sciences (EAR-1128799) and Department of Energy-GeoSciences (DE-FG02-94ER14466). This research used resources of the APS, a US Department of Energy (DOE) Office of Science User Facility operated for the DOE Office of Science by Argonne National Laboratory under contract no. DE-AC02-06CH11357.

## Author contributions

L.D., C.M. and V.C. proposed the research, did the project planning and provided the sample. E.B. and L.I. selected the single-crystals. V.C., L.D., L.I., I. Yu. K. and J.J. prepared the high-pressure experimental set-ups. V.C., L.D., E.B., M.M., L.I., M.B., V.S., S.P., M.H., C.P., V.B.P. and M.M. conducted the HPHT single-crystal X-ray diffraction experiments. V.C., L.D., C.M., I.K., A.I.C. and R.R. conducted the SMS experiments. E.B., L.D. and M.M. analysed the X-ray single-crystal diffraction data. V.C. and L.D. analysed the X-Ray powder diffraction data. V.C., C.M. and L.D. analysed the Mössbauer data. V.C., L.D. and C.M. interpreted the results and wrote the manuscript with contributions of all authors. V.C. wrote the Methods section.

## Additional information

**Competing interests:** The authors declare no competing financial interests.

