## [Peer Review File · Nature Communications]

Reviewers' Comments:

Reviewer #1 (Remarks to the Author)

This manuscript, submitted by Cerantola and co-authors, is potentially a very nice paper reporting the structures and stability of two "new" tetrahedrally co-ordinated carbonates that form from a FeCO_3 precursor at lower mantle conditions. The authors performed laser-heated diamond anvil cell experiments on synthetic single crystals of FeCO_3 at a range of PT conditions. Their results include reports of the HS-LS transition in siderite, bracketing of the melting temperature and subsequent dissociation of siderite, in-situ Mössbauer of FeCO_3 and its high PT products. These are in addition to the formation conditions and structure of $\text{Fe}_4\text{C}_3\text{O}_{12}$ and $\text{Fe}_4\text{C}_4\text{O}_{13}$ which are the "new and exciting" results of this work, however I note both that the latter carbonate is the iron end-member composition of the tetrahedral carbonate reported by Merlini et al (2015). Additionally, I believe there are some significant omissions, particularly in the methods description and of essential data, which must be provided for publication. These additions are necessary to fully assess the robustness and reliability of the results, which is especially important given the authors appear to demonstrate the structures provided in a previous paper were incorrectly refined. However, both the stability and crystal chemistry of the new phases are reasonable, so I see no reason why this should not, with revisions, become a good publication. Below I outline my suggested amendments and comments.

1. The authors report the discovery of a never-seen-before carbonate phase, with a structure that is, to the authors knowledge, different from any other compounds. The lattice parameters, structure, chemistry and stoichiometry are all determined based entirely on the refinement of a diffraction pattern collected at 97 GPa, however this pattern is not included in this submission. This makes it very hard to truly evaluate the quality of the refinement or the data - I believe this must be added before publication, possibly in supplementary information (SI).

The starting materials in experiments are single crystals of FeCO_3 , however it is unclear whether the product $\text{Fe}_4\text{C}_3\text{O}_{12}$ is also a single crystal, or a powder. The authors suggest it is a single crystal, but without a 2D diffraction image it seems more likely, especially considering the different stoichiometry of the product, to either be a powder or multiple/twinned crystals.

There is very little information provided in the Methods to indicate how the authors proceeded from a diffraction pattern to full refinement of crystal chemistry. Whilst I understand how this is feasible, some indication of the steps taken, including any assumptions made and the reasons for such assumptions should be described in detail, e.g. did the authors use the refined unit cell volumes to guide stoichiometry, how many missing reflections of the predicted reflections were there, were there any extra reflections, how did they deal with peaks from unreacted FeCO_3 starting materials etc. Some of these assumptions lead to the conclusion that all the iron in this phase is ferric, however, if this is the case why are there four components of the Mössbauer spectra in Fig S2a? Surely there should be one from FeCO_3 and two from $\text{Fe}_4\text{C}_3\text{O}_{12}$, and the spectra suggests the phase is magnetic unless I am mistaken? The authors should comment on this?

It is also notable that the quality of the refinement for $\text{Fe}_4\text{C}_3\text{O}_{12}$ is significantly worse than that for $\text{Fe}_4\text{C}_4\text{O}_{13}$ (larger GooF, R1). Why is this? This should be discussed. Also, the text says there were ~300 independent reflections, but Table S1 suggests > 1700...

Given this is the first ever report of this structure, it is rather hard to visualise the structure using fig 2 alone. Perhaps more views of the structure could be added in SI.

It is assumed that this phase is produced in addition to diamond, is there any evidence for this in the form of Debye rings in the diffraction. The authors do mention Raman spectra, but do not show them, this would be helpful.

2. The methods, in general, are extremely brief. In order to find out several pieces of key information it was necessary to look up other papers, and I remain uncertain of several points. The

authors shouldn't make it hard for readers in this way.

How was it verified that there was no temperature gradient across the sample? For instance, the normal setup on ID-27 (ESRF) uses a 2-5 micron pinhole, and therefore you cannot check this easily. Do the other setups used also collect temperatures using a pinhole, or do they use a 1D slit, collecting a profile across the whole sample? Are there examples of these profiles?

Is it known whether the starting material contain no Fe³⁺? As there is a mixed singlet/doublet spectrum at RT at 44 GPa, which could be in the starting material, or a partial transition from HS to LS.

Was recovery of the samples attempted? If so, is it known when they become amorphous/back transformed? Is this why chemical analysis using FIB, or other, wasn't attempted? Or was it attempted, as this would significantly strengthen the case for a completely new stoichiometry and chemistry.

How many independent experiments were there, and what was their individual PT paths? This particular info would be very helpful if included in a table.

How many and which diffraction patterns were single crystal, and how many were powder patterns?

The caption of fig S3 says see methods to explain something that isn't present.

3. The structural transformation sequences of the new tetrahedral carbonates are rather confusingly presented at the moment. This isn't helped by the confusing nature of Figure 1, and the poorly defined colours of the various symbols, I think there is an opacity/layering problem that is affecting the printed colours. Perhaps a key would be useful, rather than just a figure caption description? A record of the PT paths of each experiment would also be helpful, perhaps as arrows on the figures.

Are the oxide phases that form in addition to Fe₄C₄O₁₃ always consistent? It is mentioned that they can sometimes be refined from xrd, can we see an example pattern.

Line 324 suggests that FeCO₃ transforms directly to Fe₄C₄O₁₃, which is misleading because throughout other parts of the manuscript it is argued that it forms from reaction of Fe₄C₃O₁₂, so without this precursor it would not form.

Is there evidence for the growth of Fe₄C₄O₁₃ at the expense of Fe₄C₃O₁₂, such as the strengthening/weakening of diffraction peaks.

Are both products always single crystals, as the authors suggest?

Do you ever completely get rid of FeCO₃ and Fe₄C₃O₁₂ in the high T samples with Fe₄C₄O₁₃? If this was the case the final product in the DAC must contain CO₂, diamond and an Fe-oxide (e.g. Fe₃O₄) alongside the Fe₄C₄O₁₃. If there are Raman spectra is there evidence of this?

4. The order of discussion is a bit clunky. I wanted discussion of potential decomposition/formation reactions for the various phases to be much earlier, or at least nearer to the description of the results they are relevant to. I would move the discussion about fitting the data of Liu et al to after the interpretation of your own results. Whilst the discussion in 2259-281 seemed like it was out of place.

5. From reading the manuscript it is apparent that, if these tetrahedral carbonate phases are going to form in the mantle, they will have to do so initially from carbonates in cold slabs (if the reactions you propose are correct). But it is also apparent from the structural fit of Liu et al's data, that both phases form extensive solid solutions with Mg endmembers. In the case of tetrairon orthocarbonate, it is clear that Mg significantly expands its stability field at the expense of the diiron diiron tetracarbonate structure, and to higher T. This goes completely unmentioned at the moment. Clearly, given the stability fields of the various carbonates, there is a clear PT path that must be followed for these new phases to form, but once existing they might remain at geotherm temperatures. This should be discussed more, and is one of the most significant implications of this work.

6. In lines 165-183 incongruent melting is discussed. The authors, like previous authors, suggest that dissociation of carbonate produces diamond + oxides. However, in this case, the current

results clearly show this "dissociation" occurs above the melting temperature. There is no clear evidence that this is an equilibrium reaction, or whether it is a quench phenomenon related to the DAC. It is possible it is related to diffusion in a large thermal gradient, that is accelerated in the presence of melt. It is certainly clear that it is not dissociation of siderite, this should be clarified.

7. From the data provided, the values in lines 293 and 294 should be 35.9 and 13.1 respectively.

Line 44: Oceanic sediments are not the only flux of carbon into the mantle, much comes from the oceanic crust and mantle beneath.

Line 46: There is no evidence that the diamond-hosted carbonates (calcite and dolomite) reported in Kaminsky 2012 come from the lower mantle. Those carbonate inclusions are in diamonds with no other reported inclusions. Thus they categorically do not prove that carbonates exist in the lower mantle, and I therefore believe this reference is confusing. The nyerereite and nahcolite inclusions also mentioned in this paper are actually described by Kaminsky (2009, *Min Mag*) and were reported as "lower mantle" carbonates. There are other references that report lower mantle carbonate inclusions, which may or may not actually be from the LM. Several carbonate microinclusions have been reported in diamonds containing ferropericlase with magnesite exsolution (e.g. Kaminsky, *Can Mineralogist*, 2015; Kaminsky, *Mineral Petrology*, 2015); many authors are currently arguing that these must represent lower mantle samples because of magnetite exsolution observed within the inclusions, however this still seems far from clear to me. Magnesite was reported in a diamond with another inclusion interpreted as "former bridgmanite" (Thomson CTMP 2014). Also, Brenker et al (2007) (EPSL) reported carbonates in equilibrium with "former calcium perovskite", which may have been from the lower mantle.

Line 60: I don't think Rohrbach and Schmidt actually show slabs will be more oxidised than surrounding mantle.

Line 361: Kelemen and Manning 2015 actually appear to conclude, based on their figure 5, that somewhere between none and almost all of the carbonate in slabs is subducted into the mantle.

There is at least one mistake in Table S4. The final entry was at 97 GPa.

28th July 2016.

Reviewer #2 (Remarks to the Author)

In this work the authors investigated the behavior of FeCO_3 at pressures over 100 GPa and temperatures over 2500 K in a laser-heated diamond anvil cell using a combination of single crystal X-ray diffraction and Mössbauer spectroscopy. The key finding is the discovery of two new Fe-bearing carbonates, $\text{Fe}_4\text{C}_3\text{O}_{12}$ and $\text{Fe}_4\text{C}_4\text{O}_{13}$ which is isostructural with recently reported $\text{Mg}_{1.6}\text{Fe}_{2.4}\text{C}_4\text{O}_{13}$ by Merlini et al. (2015). The latter observation suggests that ca. 35% of magnesium may be incorporated in the structure and that this structure may be an important phase to consider in the Earth's deep interior. The authors conclude that the conditions in the Earth's lower mantle may not lead to full decomposition of Fe-based carbonates - at least over the time-range of the experiments performed.

While the authors attempt to relate these and other findings to the stability of iron-bearing carbonates in the Earth's deep interior, I found that the manuscript deviated from this theme and lost its coherency. I suggest, for example, that the details of structure and crystal chemistry of the newly-discovered iron carbonates be published in another manuscript. The details of the spin transition do not need a separate section. The manuscript should focus on the stability and the thermodynamic conditions of the decarbonation process in magnesium and iron-bearing carbonates. However, while this manuscript provides intriguing new details, this remains poorly constrained as seen in Figure 1 which attempts to compile all known data. The reactions depend

critically upon the redox conditions and a further discussion of this subject is warranted. Thus, in its existing form, I cannot recommend that the manuscript be accepted for publication.

**Reviewer #1**

“This manuscript, submitted by Cerantola and co-authors, is potentially a very nice paper
reporting the structures and stability of two "new" tetrahedrally co-ordinated carbonates that
form from a FeCO₃ precursor at lower mantle conditions. The authors performed laser-heated
diamond anvil cell experiments on synthetic single crystals of FeCO₃ at a range of PT
conditions. Their results include reports of the HS-LS transition in siderite, bracketing of the
melting temperature and subsequent dissociation of siderite, in-situ Mössbauer of FeCO₃ and its
high PT products.“

We appreciate that Reviewer #1 recognized the potential significance of our results.

“These are in addition to the formation conditions and structure of Fe₄C₃O₁₂ and Fe₄C₄O₁₃
which are the "new and exciting" results of this work, however I note both that the latter
carbonate is the iron end-member composition of the tetrahedral carbonate reported by Merlini et
al (2015).”

We clearly stated in the paper that Fe₄C₄O₁₃ is isostructural with Mg₂Fe₂C₄O₁₃, and we provide
appropriate reference (ref. 23) (and we also explain why we consider the synthesis of pure iron
compound Fe₄C₄O₁₃ important). Fe₄C₃O₁₂ on the other hand was synthesized by us for the first
time. As Reviewer #1 certainly noticed, Dr. Marco Merlini is co-author of the present work, and
we are sure he never published findings of the same phase before.

“ The authors report the discovery of a never-seen-before carbonate phase, with a structure that
is, to the authors knowledge, different from any other compounds. The lattice parameters,
structure, chemistry and stoichiometry are all determined based entirely on the refinement of a
diffraction pattern collected at 97 GPa, however this pattern is not included in this submission.
This makes it very hard to truly evaluate the quality of the refinement or the data - I believe this
must be added before publication, possibly in supplementary information (SI).”

The structures of these novel iron carbonates were determined not from powder, but from single
crystal diffraction data. We provide now as supplements CIF files which contain all
crystallographic information as required by IUCr.

“The starting materials in experiments are single crystals of FeCO₃, however it is unclear
whether the product Fe₄C₃O₁₂ is also a single crystal, or a powder. The authors suggest it is a
single crystal, but without a 2D diffraction image it seems more likely, especially considering the
different stoichiometry of the product, to either be a powder or multiple/twinned crystals.”

We thank the reviewer to highlight this point. All our structural refinements have been
performed on single crystal domains that we identified in the diffraction patterns, see for
example the 2D image, Fig. S1 in SI (added in the revised version). For instance, we refined the
structure of Fe₄C₃O₁₂ at ~74 GPa after heating at ~1750 K, choosing one out of 4 identified
single crystal domains. This does not mean we did not observe the signature typical for powder
diffraction, however we could identify for all reported data single crystal domains that allowed
45 us to perform single crystal structural refinement.

There is very little information provided in the Methods to indicate how the authors proceeded
from a diffraction pattern to full refinement of crystal chemistry. Whilst I understand how this is
feasible, some indication of the steps taken, including any assumptions made and the reasons for
such assumptions should be described in detail, e.g. did the authors use the refined unit cell
volumes to guide stoichiometry, how many missing reflections of the predicted reflections were
there, were there any extra reflections, how did they deal with peaks from unreacted FeCO₃
starting materials etc.

It is also notable that the quality of the refinement for Fe₄C₃O₁₂ is significantly worse than that
for Fe₄C₄O₁₃ (larger GooF, R1). Why is this? This should be discussed. Also, the text says
there were ~300 independent reflections, but Table S1 suggests > 1700...”

We appreciate Reviewer #1 advise to include in the paper more methodological and technical
details how structural analysis was performed. Fortunately, methodology of single crystal
structural analysis at high pressure (and high temperatures) was rapidly developed over last few
61 years, and the way “how the authors proceeded from a diffraction pattern to full refinement of
62 crystal chemistry” is described in numerous publications of some of co-authors of current work
and other colleagues (see, for example, refs. 8,16,23,26,27,39,48,), accumulated in special issues
(particularly, High Pressure Research, 33, 2013), and in PhD Thesis (see, for example,
<https://epub.uni-bayreuth.de/2124/>). It seems not feasible to summarize all these technological
and computational procedures in few sentences. However, the method is currently applied by
several groups, and fully established.

We supplement now the manuscript with CIF files (checked by UICr CIFchecker) which is
supposed to address Reviewer's #1 concerns regarding assumptions, number of reflections used,
etc. Please note that Table S2 (former Table S1, which present some data contained in CIFs)
shows the absence of any constrains during structural solution and refinement (it also means that
no assumptions were made to refine chemical composition), number of observables are 6 to 10
73 times larger than number of determined parameters, and we have quite high redundancy factors
(i.e. number of observed reflection is 4.5-5 times higher than independent one).

Using additional experimental data and further analysis we improved quality of structural
refinement of $\text{Fe}_4\text{C}_3\text{O}_{12}$ as shown in corresponding tables and CIF.

Reviewer #1 is absolutely correct, upon laser heating we get several single crystal domains of the
same phase (and we now state it explicitly in the text), but modern crystallographic software
(particularly CrysAlisPro which we are using) can handle this specific cases.

“Some of these assumptions lead to the conclusion that all the iron in this phase is ferric,
however, if this is the case why are there four components of the mössbauer spectra in Fig S2a?
Surely there should be one from FeCO_3 and two from $\text{Fe}_4\text{C}_3\text{O}_{12}$, and the spectra suggests the
phase is magnetic unless I am mistaken? The authors should comment on this?”

As we explain above (and as it is also stated in the paper) the chemical composition of $\text{Fe}_4\text{C}_3\text{O}_{12}$
was determined directly by solving the structure of the material from single crystal X-ray
diffraction data, without any additional assumption. Conclusion that all iron is ferric is made
based on simple count.

As we explained in the text, the “accurate Mössbauer spectroscopy characterisation of the pure
phase is difficult due to presence of other iron compounds in the laser-heated samples”. For this
reason it is not trivial to address the magnetic components observed in the spectrum. The intense
magnetic sextet(s) could represent the two Fe-sites in $\text{Fe}_4\text{C}_3\text{O}_{12}$, because we did not observe any
magnetic contribution in the Mössbauer spectrum of $\text{Fe}_4\text{C}_4\text{O}_{13}$ + Fe-oxides that we reported in
Fig. S2b. However, since we could not isolate the orthocarbonate from the other phases we
prefer to do not speculate on the magnetism of this phase for now. We would like also mention
(and this is stated in the text as well) that all components in Mössbauer spectra collected on the
samples containing HP-carbonate phases belongs to iron in high-spin state, and this we consider
as important (and quit robust) result.

“Given this is the first ever report of this structure, it is rather hard to visualize the structure
using fig 2 alone. Perhaps more views of the structure could be added in SI.”

As Reviewer #1 proposes we add a new Fig. 2 with other views of the structures. Moreover, with
this revised version of the manuscript we provide CIF files, so drawing and visualization of the
crystal structure become immediate. Also, with slightly more effort, Table S2 (former Table S1)
contains all necessary information to draw models of crystal structures.

“It is assumed that this phase is produced in addition to diamond, is there any evidence for this in
the form of debye rings in the diffraction. The authors do mention Raman spectra, but do not
show them, this would be helpful.”

It seem to be not feasible to identify unambiguously diamond from weak powder diffraction lines
in very complex environment of strongly scattering iron-bearing phases. As Reviewer #1
proposes we added Raman spectra in SI (Fig. S3).

“ The methods, in general, are extremely brief. In order to find out several pieces of key
information it was necessary to look up other papers, and I remain uncertain of several points.
The authors shouldn't make it hard for readers in this way.

We fully agree with Reviewer #1 and often we experience the same problems reading papers in
journals of Nature series. However, we have to obey Nature Communications rules which stay
“*Methods should be written as concisely as possible*”. So, we did our best to provide “*all*
*elements necessary for interpretation and replication of the results*”. We thank Reviewer #1 for
pointing some weaknesses and try to address concerns as best as we can.

“How was it verified that there was no temperature gradient across the sample? For instance, the
normal setup on ID-27 (ESRF) uses a 2-5 micron pinhole, and therefore you cannot check this
easily. Do the other setups used also collect temperatures using a pinhole, or do they use a 1D
slit, collecting a profile across the whole sample? Are there examples of these profiles?”

Temperature gradients in LH-DAC experiments are often present. They can be generated by
misalignment between laser and X-ray beam, as well as by difference in sample absorption due
to changing sample thickness or surface roughness. Using a small X-ray beam with respect to the
laser-heating beam can prevent these situations, proven that they are well align with respect to
each other and the sample size is smaller than the laser beam hot spot. In all our experiments we
fulfilled these requirements and we feel safe to conclude that in within the measured area we did
not have significant (higher than the reported experimental uncertainties) temperature gradients.
Moreover, in revised version of the manuscript we explicitly state “*Methodological aspects of*
*the high pressure experiments on these beam-lines are well establishe*^{8,16,2326,48}”. We explain in
the manuscript that “Crystals (as a rule about 10 μm in diameter) were completely ‘surrounded’
by laser light and there were no measurable temperature gradients within the samples”.

The laser heating set-ups at both ID27 (ESRF) and IDD-13 (APS) allow collecting temperature
profile through the heated spot, and we have examples of such profiles. There are already many
examples of such pictures in publications of our and other groups, and we do not think that
adding one more profile (out of hundreds measurements performed during laser heating in this
particular work) could contribute to justify the quality of experiments.

“Is it known whether the starting material contain no Fe³⁺? As there is a mixed singlet/doublet
spectrum at RT at 44 GPa, which could be in the starting material, or a partial transition from HS
to LS.”

The starting material has no Fe³⁺. The presence of only Fe²⁺ has been confirmed by Mossbauer
spectroscopy (Cerantola et al.¹⁹) and by single crystal diffraction. In fact, the question raised by
Reviewer #1 is addressed in Ref. 19 and we explicitly refer to it.

As reported in the caption of Fig. 4 (former Fig. 3), the spectra at 45 GPa have been collected
before (left) and during heating (right) at 570(50) K. The blue doublet corresponds to the high
spin of ferrous iron whereas the red singlet stands for low spin state. The intensity (amount) of
high spin component increases with increasing temperature.

“Was recovery of the samples attempted? If so, is it known when they become amorphous/back
transformed? Is this why chemical analysis using FIB, or other, wasn't attempted? Or was it
attempted, as this would significantly strengthen the case for a completely new stoichiometry and
chemistry.”

We undertake few attempts to FIB recovered samples, but it proven to be difficult for small
laser-heated crystals and material was lost (note we are not dealing with compact samples
usually recovered after powder diffraction experiments). We continuously try to get TEM data
on recovered materials, but we consider this part of work independent from that we are
presenting here and, in fact, of secondary priority: (a) we obtained data about chemistry and
structure of materials in situ by employing single crystal diffraction analysis (and Mössbauer
spectroscopy as well), and (b) comparison of our results with literature reports show that TEM
analyses of the recovered samples (even in combination with in situ powder diffraction) cannot
provide the full picture of the chemical processes (for example, not all carbonate and oxide
phases were identified) and give information about the crystal structures of the high pressure
materials.

“How many independent experiments were there, and what was their individual PT paths? This
particular info would be very helpful if included in a table.

How many and which diffraction patterns were single crystal, and how many were powder
patterns?”

We provide new Table S1 (in SI) which contains information about all data-points shown on Fig.
1.

“The caption of fig S3 says see methods to explain something that isn't present.”

We corrected figures captions and Methods. All necessary references (particularly, #50, #51,
#52, #53, #54 and #56) are provided.

“The structural transformation sequences of the new tetrahedral carbonates are rather
confusingly presented at the moment. This isn't helped by the confusing nature of Figure 1, and
the poorly defined colours of the various symbols, I think there is an opacity/layering problem
that is affecting the printed colours. Perhaps a key would be useful, rather than just a figure
caption description? A record of the PT paths of each experiment would also be helpful, perhaps
as arrows on the figures.”

We did our best to improve the presentation; particularly, Table S1 is now added, and Fig. 1 is
modified taking in to account Reviewer #1 suggestions.

“Are the oxide phases that form in addition to Fe₄C₄O₁₃ always consistent? It is mentioned that
they can sometimes be refined from xrd, can we see an example pattern.”

The 2D image containing diffraction rings of HP-Fe₃O₄ is shown in Fig. 3. In Table S5 (former
Table S3) we provide examples of the results of single crystal structural refinements of iron
oxides we found in experiment with iron carbonate – Fe₁₃O₁₉, HP-Fe₃O₄, Fe₅O₇, and ppv-Fe₂O₃.
We report lattice parameters, space groups, number of detected reflections (range from about 160
to over 400), R-factors, etc. Our results can be directly compared with data from ISCD data-base
(for reader's convenience ref numbers are also provided), and thus phases identification is
unambiguous.

Line 324 suggests that FeCO₃ transforms directly to Fe₄C₄O₁₃, which is misleading because
throughout other parts of the manuscript it is argued that it forms from reaction of Fe₄C₃O₁₂, so
without this precursor it would not form.

We did our best to avoid any definite statements if $\text{Fe}_4\text{C}_4\text{O}_{13}$ may be obtained directly from
FeCO_3 . In particular, we wrote: “Laser heating of FeCO_3 at temperatures above 1750(100) K at
pressures above ~ 74 GPa resulted in formation not only $\text{Fe}_4\text{C}_3\text{O}_{12}$ and iron oxides (see below)
but also a monoclinic (space group $C2/c$, #15) phase”. Later we explain: “The monoclinic
*diiron(II) diiron(III) tetracarbonate*, $\text{Fe}_4\text{C}_4\text{O}_{13}$ appear only upon prolonged (about one hour)
laser heating above 1650(100) K”. In the Discussion section we consider different scenario
which could lead to formation of $\text{Fe}_4\text{C}_4\text{O}_{13}$, and based on observations propose that it forms
through decomposition of $\text{Fe}_4\text{C}_3\text{O}_{12}$. Experimental observations supporting our idea are
presented on Fig. 5, where the progressive heating of FeCO_3 at different temperatures and
constant pressure (~ 110 GPa) shows the transformation sequence: $\text{FeCO}_3 \rightarrow \text{Fe}_4\text{C}_3\text{O}_{12} \rightarrow$
$\text{Fe}_4\text{C}_4\text{O}_{13}$.

Is there evidence for the growth of \$\text{Fe}_4\text{C}_4\text{O}_{13}\$ at the expense of \$\text{Fe}_4\text{C}_3\text{O}_{12}\$, such as the
strengthening/weakening of diffraction peaks.

We obviously see some changes in relative intensities of the reflections, but in the complex
diffraction patterns from mixtures containing several components (some powdered and some
highly spotty polycrystalline) it would be too speculative to draw any conclusions based on such
observations.

Are both products always single crystals, as the authors suggest?

Yes they are. But of course we observed powdered (or highly textured polycrystalline) patterns
as well. Examples are given i.e. in Fig. 3 (former Fig. 4). See also the new Fig. S1 (added in the
revised version) with the example of FeCO_3 transformation to $\text{Fe}_4\text{C}_3\text{O}_{12}$ and observation of new
diffraction spots.

Do you ever completely get rid of \$\text{FeCO}_3\$ and \$\text{Fe}_4\text{C}_3\text{O}_{12}\$ in the high T samples with \$\text{Fe}_4\text{C}_4\text{O}_{13}\$?
If this was the case the final product in the DAC must contain \$\text{CO}_2\$, diamond and an Fe-oxide
(e.g. \$\text{Fe}_3\text{O}_4\$ ) alongside the \$\text{Fe}_4\text{C}_4\text{O}_{13}\$. If there are Raman spectra is there evidence of this?”

In Fig. 1 we show a region of coexistence between $\text{Fe}_4\text{C}_3\text{O}_{12}$ and $\text{Fe}_4\text{C}_4\text{O}_{13}$. However, we have
the case when samples treated at pressures about 100 GPa and temperatures above 2500 K
contain $\text{Fe}_4\text{C}_4\text{O}_{13}$ and iron oxide(s) (Fe_3O_4 and Fe_2O_3). We did not observe CO_2 (and did not
expect to observe because it is known that CO_2 is not stable at such conditions). We could not
find evidences for presence of pure oxygen by diffraction, which is not so surprising – oxygen is
low-Z material in comparison with iron-bearing compounds, and may also dissolve/mix with Ne
pressure medium. As we mentioned in the text (and as it should be evident from Fig. S3 (which

is added in this revised version), Raman spectroscopy in the presence of nanocrystalline diamond
(and probably other highly fluorescing products of laser heating experiments) has no chances to
spot oxygen.

“The order of discussion is a bit clunky. I wanted discussion of potential
decomposition/formation reactions for the various phases to be much earlier, or at least nearer to
the description of the results they are relevant to. I would move the discussion about fitting the
data of Liu et al to after the interpretation of your own results. Whilst the discussion in 2259-281
seemed like it was out of place.”

We thank the reviewer for the advise. As Reviewer #1 suggests we re-arrange the Discussion
section, and particularly we wrote a new sub-section which dedicated to incorporation of Mg in
CO₄-bearing carbonates.

“From reading the manuscript it is apparent that, if these tetrahedral carbonate phases are going
to form in the mantle, they will have to do so initially from carbonates in cold slabs (if the
reactions you propose are correct). But it is also apparent from the structural fit of Liu et al's
data, that both phases form extensive solid solutions with Mg endmembers. In the case of
tetrairon orthocarbonate, it is clear that Mg significantly expands its stability field at the expense
of the diiron diiron tetracarbonate structure, and to higher T. This goes completely unmentioned
at the moment. Clearly, given the stability fields of the various carbonates, there is a clear PT
path that must be followed for these new phases to form, but once existing they might remain at
geotherm temperatures. This should be discussed more, and is one of the most significant
implications of this work.”

We appreciate Reviewer #1 advise to discuss possible stability of Mg,Fe-orthocarbonate along
the geotherm, and we followed this suggestion. However, we also feel that unless solid
experimental evidences are at hands the idea should be formulated with caution.

“In lines 165-183 incongruent melting is discussed. The authors, like previous authors, suggest
that dissociation of carbonate produces diamond + oxides. However, in this case, the current
results clearly show this "dissociation" occurs above the melting temperature. There is no clear
evidence that this is an equilibrium reaction, or whether it is a quench phenomenon related to the
DAC. It is possible it is related to diffusion in a large thermal gradient, that is accelerated in the
presence of melt. It is certainly clear that it is not dissociation of siderite, this should be
clarified.”

We agree with the reviewer that this might not be an equilibrium reaction. Indeed we cannot
exclude that after prolonged heating (several hours, days?) FeCO_3 will completely decompose to
form $\alpha\text{-Fe}_2\text{O}_3$ or $\text{HP-Fe}_3\text{O}_4$. However, we earnestly described our results. Based on what we
observed there are no evidences that dissociation of siderite is an artifact related to laser heating
in DACs (at least because results in DACs and multi-anvil apparatuses are essentially the same).

“From the data provided, the values in lines 293 and 294 should be 35.9 and 13.1 respectively.”

Corrected.

“Line 44: Oceanic sediments are not the only flux of carbon into the mantle, much comes from
the oceanic crust and mantle beneath.”

Corrected.

“Line 46: There is no evidence that the diamond-hosted carbonates (calcite and dolomite)
reported in Kaminsky 2012 come from the lower mantle. Those carbonate inclusions are in
diamonds with no other reported inclusions. Thus they categorically do not prove that carbonates
exist in the lower mantle, and I therefore believe this reference is confusing. The nyerereite and
nahcolite inclusions also mentioned in this paper are actually described by Kaminsky (2009, Min
Mag) and were reported as "lower mantle" carbonates. There are other references that report
lower mantle carbonate inclusions, which may or may not actually be from the LM. Several
carbonate microinclusions have been reported in diamonds containing ferropericlase with
magnesite exsolution (e.g. Kaminsky, Can Mineralogist, 2015; Kaminsky, Mineral Petrology,
2015); many authors are currently arguing that these must represent lower mantle samples
because of magnetite exsolution observed within the inclusions, however this still seems far from
clear to me. Magnesite was reported in a diamond with another inclusion interpreted as "former
bridgmanite" (Thomson CTMP 2014). Also, Brenker et al (2007) (EPSL) reported carbonates in
equilibrium with "former calcium perovskite", which may have been from the lower mantle.”

We “soften” formulations and provide new references as Reviewer #1 suggests (refs. 6 and 7).

“Line 60: I don't think Rohrbach and Schmidt actually show slabs will be more oxidised than
surrounding mantle.”

Rohrbach and Schmidt¹⁴ describe two mechanisms, redox freezing and redox melting, where
carbonatite melts reduce and form diamonds (redox freezing) or diamonds oxidize forming CO₂
(redox melting). Both processes might happen at the 660 Km-discontinuity where the subducting
lithosphere deflects into the transition zone or stagnate into the lower mantle (redox freezing) or
during upwelling, when the lower mantle constituted mainly by Fe³⁺-rich perovskite oxidizes
diamonds due to the sudden increase in Fe³⁺ activity, which is not counterbalanced by an
adequate amount for instance of metal iron (redox melting).

In their manuscript, Rohrbach and Schmidt¹⁴ write: “ **Starting from a subducting, locally**
**carbonated relatively oxidized mafic to ultramafic lithosphere**, our experiments demonstrate
that carbonatite melts will be generated in such lithosphere on thermal relaxation”.

“Kelemen and Manning 2015 actually appear to conclude, based on their figure 5, that
somewhere between none and almost all of the carbonate in slabs is subducted into the mantle.”

We appreciate Reviewer #1 comment and incorporate it in the text.

“There is at least one mistake in Table S4. The final entry was at 97 GPa.”

We thank Reviewer #1 for pointing it out, but this is not a mistake – at the moment of data
collection of SMS pressure in the cell was 92(2) GPa.

**Reviewer #2**

“In this work the authors investigated the behavior of FeCO₃ at pressures over 100 GPa and
temperatures over 2500 K in a laser-heated diamond anvil cell using a combination of single
crystal X-ray diffraction and Mössbauer spectroscopy. The key finding is the discovery of two
new Fe-bearing carbonates, Fe₄C₃O₁₂ and Fe₄C₄O₁₃ which is isostructural with recently
reported Mg_{1.6}Fe_{2.4}C₄O₁₃ by Merlini et al. (2015). The latter observation suggests that ca.
35% of magnesium may be incorporated in the structure and that this structure may be an
important phase to consider in the Earth’s deep interior. The authors conclude that the conditions

in the Earth's lower mantle may not lead to full decomposition of Fe-based carbonates - at least
over the time-range of the experiments performed.

While the authors attempt to relate these and other findings to the stability of iron-bearing
carbonates in the Earth's deep interior, I found that the manuscript deviated from this theme and
lost its coherency. I suggest, for example, that the details of structure and crystal chemistry of the
newly-discovered iron carbonates be published in another manuscript.”

We thank Reviewer #2 for the comments, however we do not agree with Reviewer #2 judgment
that manuscript deviate from declared subject – stability of iron-bearing carbonates at conditions
of deep Earth interiors. In fact, we would argue our work clarify the most fundamental question –
could iron carbonates sustain pressure-temperature conditions of lower mantle, and if “yes” in
which form they may exist. We demonstrate that self-redox reaction does not lead to the break
down of carbonates to oxides (or elements), but instead results in the formation of more complex
(and unexpected) compounds. We are fully convinced that without unambiguous identification of
the products of the reactions and their phase/structural characterization any possible analysis of
the fate of iron-bearing carbonates in the Earth interior would be just speculations. We are glad
that Editor explicitly “recommend do keep” “presentation of the new crystal chemistry”.

The details of the spin transition do not need a separate section.

As Reviewer #2 proposed we eliminate separate section on the spin crossover.

“The manuscript should focus on the stability and the thermodynamic conditions of the
decarbonation process in magnesium and iron-bearing carbonates. However, while this
manuscript provides intriguing new details, this remains poorly constrained as seen in Figure 1
which attempts to compile all known data. “

We do not understand why Reviewer #2 believes we have to focus on “decarbonation process”.
In fact, instead of “decarbonation” we observed formation of novel carbonates. This result is
unambiguous and clearly reflected on Fig. 1.

“The reactions depend critically upon the redox conditions and a further discussion of this
subject is warranted.”

We agree with Reviewer #2 statement. In the revised version of the manuscript we explain that
while only reaction (5) explicitly depends on oxygen fugacity, all other processes described in

section “*Chemical transformations of FeCO₃ at high pressures and temperatures*” may be
affected by redox conditions (or play role of buffering reactions in more complex process
involving iron-bearing carbonates). Moreover, in the view of the recently reported (see for
example Refs. 26,42) fundamental changes in the chemical behaviour of iron-oxygen system at
pressures above ~70 GPa and high temperatures, our results are also calling for detail
investigations of redox processes in lower part of Earth lower mantle and core-mantle boundary.
Here we provide important new findings and information but we perfectly understand that the
effort of one group is not sufficient to fully resolve the problem, and we hope that publication of
our work will promote further studies in the direction also proposed by Reviewer #2.

Reviewers' Comments:

Reviewer #1:

Remarks to the Author:

I have read the revised manuscript and replies to the reviewers' comments, and have found the response convincing for the most part, and the authors have certainly managed to address my technical concerns. The key results of Cerantola et al. manuscript can be summarised as:

- The identification and structure determination of $\text{Fe}_4\text{C}_3\text{O}_{12}$, the iron endmember of the high pressure carbonate structure reported by Merlini et al (2015).
- The identification and structure determination of $\text{Fe}_3\text{C}_3\text{O}_{12}$ – a new high pressure carbonate mineral constructed from CO_4 tetrahedra. It has also been confirmed that "siderite-II" from previous studies likely was this structure.
- That these tetrahedrally coordinated iron carbonates are stable at adiabatic temperatures. Combined with their ability to form partial solid solutions with magnesium-bearing compositions suggests they might exist in the Earth's lower mantle.

The new manuscript appears much improved, and presents a more concise and coherent report of the results. I retain the following issues, which need not necessarily be an impediment to publication of the study:

- In my opinion the authors still do not present a convincing argument demonstrating that there is a mechanism that would allow delivery of oxidised carbon into the deep mantle and thus, do not demonstrate that these new carbonate structures should be expected to be part of the lower mantle assemblage.
- It is not clear whether these structures are stable in mantle bulk compositions, which are much more magnesium-rich even than the study of Merlini et al. (2015). The relevance of endmember iron carbonates is not entirely clear.
- The interpretation of Mössbauer spectra remains confusing. In my opinion, the lack of certainty surrounding this aspect of the manuscript acts to raise doubt, rather than support the key arguments.

Reviewer #2:

Remarks to the Author:

The authors have addressed all of the reviewers comments.

The revised manuscript is acceptable for publication in Nature.

**Reviewer #1**

“I have read the revised manuscript and replies to the reviewers’ comments, and have found the
response convincing for the most part, and the authors have certainly managed to address my
technical concerns.

...

The new manuscript appears much improved, and presents a more concise and coherent report of
the results.”

We appreciate Reviewer #1 recognized our effort to improve the manuscript following his/her
advises. We thank Reviewer #1 for supporting the publication of our manuscript.

“I retain the following issues, which need not necessarily be an impediment to publication of the
study:

- In my opinion the authors still do not present a convincing argument demonstrating that there is
a mechanism that would allow delivery of oxidised carbon into the deep mantle and thus, do not
demonstrate that these new carbonate structures should be expected to be part of the lower
mantle assemblage”

Presence of carbonates in lower mantle is not a hypothesis – it is proven by the finding of
carbonates inclusions in diamonds (see, for example, Kaminsky F. (2012) Mineralogy of the
lower mantle: a review of 'superdeep' mineral inclusions in diamond. Earth Sci. Rev. 110. 127–
147, now added as a reference in the main text, Ref. 8) In our manuscript we report two new
structures characterized by C-O tetramers with different degrees of polymerization (ortho- and
tetracarbonate structures), and clearly demonstrate their stability at lower mantle conditions.
Thus we provide evidence that certain form of carbonates may be present in Earth lower mantle.
Uncover mechanism of their delivery to deep interiors or primordial nature is behind scope of the
work.

“It is not clear whether these structures are stable in mantle bulk compositions, which are much
more magnesium-rich even than the study of Merlini et al. (2015). The relevance of endmember
iron carbonates is not entirely clear.”

We thank Reviewer #1 to point this out, indeed we are aware of this. We performed this study
starting from the FeCO₃ endmember in order to investigate the ultimate case of possible
destabilization of iron-bearing carbonates due to self-redox reaction(s) at conditions of Earth
lower mantle. We found that instead of full decomposition, iron carbonate formed novel

compounds and thus we address the stability of Fe-carbonates. Of course, the next natural step is
to study behavior of Mg-rich materials, and it is in focus of our research now.

“The interpretation of mössbauer spectra remains confusing. In my opinion, the lack of certainty
surrounding this aspect of the manuscript acts to raise doubt, rather than support the key
arguments.”

We believe Reviewer #1 refers to our attempt to fit the Mössbauer spectrum of the HP-
carbonates phases. We fully agree with Reviewer #1 that analysis of very complex Mössbauer
spectra reported in Fig. S2 is very difficult due to overlap of components from different phases
and different iron species. Still, the data serve its purpose – assignment of possible oxidation
states of iron in laser-heated samples. We are convinced that at the moment there is no any way
to provide better resolved spectra, or spectra of pure phases, at conditions of our experiments.
Thus, we believe that solid experimental data should be reported, with the intent as well to
provide basis for further studies/analysis of different groups.

**Reviewer #2**

“The authors have addressed all of the reviewers comments.

The revised manuscript is acceptable for publication in Nature.”

We thank Reviewer #2 for his support to publish our manuscript in Nature Communications.
